# A Two-Layer Controller for Lateral Path Tracking Control of Autonomous Vehicles

**DOI:** 10.3390/s20133689

**Published:** 2020-07-01

**Authors:** Zhiwei He, Linzhen Nie, Zhishuai Yin, Song Huang

**Affiliations:** 1School of Automotive Engineering, Wuhan University of Technology, Wuhan 430070, China; whlgcjhhzw14@whut.edu.cn (Z.H.); linzhen_nie@whut.edu.cn (L.N.); huangs@dfmc.com.cn (S.H.); 2Hubei Key Laboratory of Advanced Technology for Automotive Components, Wuhan University of Technology, Wuhan 430070, China

**Keywords:** autonomous vehicle, lateral path tracking control, model predictive control

## Abstract

This paper presents a two-layer controller for accurate and robust lateral path tracking control of highly automated vehicles. The upper-layer controller, which produces the front wheel steering angle, is implemented with a Linear Time-Varying MPC (LTV-MPC) whose prediction and control horizon are both optimized offline with particle swarm optimization (PSO) under varying working conditions. A constraint on the slip angle is imposed to prevent lateral forces from saturation to guarantee vehicle stability. The lower layer is a radial basis function neural network proportion-integral-derivative (RBFNN-PID) controller that generates electric current control signals executable by the steering motor to rapidly track the target steering angle. The nonlinear characteristics of the steering system are modeled and are identified on-line with the RBFNN so that the PID controller’s control parameters can be adjusted adaptively. The results of CarSim-Matlab/Simulink joint simulations show that the proposed hierarchical controller achieves a good level of path tracking accuracy while maintaining vehicle stability throughout the path tracking process, and is robust to dynamic changes in vehicle velocities and road adhesion coefficients.

## 1. Introduction

Despite significant advances made in recent years, highly or fully automated driving of vehicles remains challenging in arbitrarily complex environments, due to numerous non-trivial issues to be addressed, among which is the path tracking control [1]. The aim of designing the path tracking controller is to ensure that the vehicle follows reference paths accurately and robustly in a timely manner under varying environmental and vehicular conditions on the premise of guaranteed vehicle stability [2]. In this study, we focus on the lateral control of autonomous vehicles.

Previously proposed path tracking schemes can be classified into 3 categories: (1) Geometric based control, including pure pursuit control [3,4], the Stanley Tracking Algorithm [5], etc., with which the front wheel angle is computed by investigating the geometric relationship among vehicle kinematics, reference paths and points of preview. Due to the neglect of vehicle dynamics in these strategies, both tracking accuracy and vehicle stability would be worsened as the vehicle speed increases. (2) Feedback control without prediction, including backstepping control [6], *H*_∞_ control [7], sliding-mode control [8,9], adaptive control [10,11] and the combination of aforementioned algorithms [12]. In these strategies, explicit control laws are designed with regard to vehicle dynamics whose key characteristics are captured with mathematically complex models. Although it has been proven by both simulation studies and field tests that these control algorithms are effective in vehicle motion control, their performance under highly dynamic conditions are not satisfactory mainly due to the absence of prediction on future conditions and low tolerance on external disturbances [13]. (3) Model predictive control (MPC), which is well suited to optimization problems with multiple constraints and is characterized by prediction and optimization, is now increasingly adopted in implementing path tracking control of autonomous vehicles [14,15]. A path tracking controller built in the MPC framework employs a vehicle dynamics model to predict vehicle states and establishes a multivariate multi-objective function between the predicted states of the vehicle and the reference variables as an open-loop optimal problem. At each sampling time, a sequence of optimal steering angles is calculated by solving the control problem with multiple constraints, which is applied to the control process only during the following sampling interval. Then, a new optimal control sequence is calculated based on new measurements of the vehicle states over a shifted horizon. The benefit brought by rolling optimization on finite and shifting horizons is that MPC can optimize its control law throughout the process of control and therefore can cope with the dynamically changing characteristics of the controlled system. The linear time-varying (LTV) MPC is more applicable in practice as the basic control strategy due to its much higher computational efficiency as compared to nonlinear MPC (NMPC) [16].

Full capture of the nonlinearities of the vehicle in the prediction model is neither favorable nor possible when designing MPC controllers due to the enormously large computation overhead [17]. As a result, extensive efforts have been invested in proposing linearized models that can contribute to achieving a tradeoff between tracking accuracy and computation efficiency. Among these proposals, the proportional linearized tire model with the small-angle assumption is one that gets widely adopted in previously proposed MPC-based controllers [18,19,20]. However, it is not necessarily true to assume that the slip angle will stay small, especially when the vehicle speed is high and the road friction coefficient is low. Significant modeling error could be caused if the linearized tire model is used without imposing constraints on the slip angle as the lateral tire forces will be saturated and no longer increase linearly with the slip angle when the lateral vehicle dynamics enter the nonlinear zone. As a result, the predicted states of the vehicle would deviate from actual values and therefore affect control accuracy [21]. Although adopting a nonlinear tire model contributes to minimizing the modeling error, it is still important for the slip angle to be contained within a small interval. In the framework of LTV-MPC, the nonlinear prediction model is linearized at each control point of operation. As a result, even with a nonlinear tire model, significant modeling errors would still be caused when the control points of operation approach the nonlinear zone of tire dynamics. Therefore, a constraint on the slip angle is required.

To develop a path tracking controller that is robust to dynamic changes in working conditions and yet still benefits from the low computation burden with LTV-MPC, a nonlinear tire model with a constraint imposed on the slip angle is incorporated into the prediction model in this study.

To cope with arbitrarily complex driving conditions, a few proposals have also focused on dynamic control parameters including optimal sampling time [22,23] or adaptive weights of the cost function [15]. In [24], the authors point out the necessity of changing both the prediction horizon and control horizon as speed changes to maintain vehicle stability. Though, there is still limited research on fine-tuning the prediction horizon and control horizon other than setting these two parameters empirically or via trial-and-error [25,26]. This study proposes to optimize both parameters with regard to various vehicle speeds and road adhesion coefficients using the particle swarm optimization (PSO) algorithm.

Another major gap between previously proposed path tracking controllers and practical lateral control of autonomous vehicles is the absence of a precise model that depicts the nonlinear characteristics of the steering system. The consequences are two-fold.

First, the feedback of the actual steering angle is almost certainly distorted as the nonlinearities of the steering system are either completely neglected or overly simplified [27]. Some efforts have been made to approximate the steering system with linear representations. In [28], the steering system is identified with the prediction-error minimization (PEM) method, and a second-order system is established as the transfer function between the actual steering angle and the target. Han at al. [10] designed a second-order steering control model with the control auto-regression and moving-average (CARMA) method, and the parameters of the steering system are estimated with the method of forgetting factor recursive least squares (FFRLS). Though, the linear approximation of the rather complex nonlinear time-varying steering systems is deemed to be non-applicable for practical purposes. Besides, identifying system parameters under different steering conditions is quite tedious.

Second, as the steering angle is not a physical signal applicable to any type of steering actuators, it becomes impossible to verify the effectiveness of proposed controlling strategies via field tests.

Therefore, an efficient and robust control strategy developed based on an accurate steering system model, that outputs executable control signals such as electric current by actuators is an important step toward developing path tracking controllers for real-life autonomous driving. To identify the nonlinear characteristics of the controlled system online and adjust the control parameters adaptively, adaptive PID control strategies based on neural networks have been developed [10]. Li at al. [29] proposed a back propagation neural network (BPNN)-PID control strategy to eliminate the nonlinear friction in the electric power steering system. In our case, a RBFNN, which converges faster as compared to BPNN, is adopted.

This paper proposes a two-layer path tracking controller for lateral control of autonomous vehicles. The upper layer is a PSO-LTV-MPC controller built upon a 3-DOF vehicle dynamics model. With the root mean square (RMS) value of tracking deviation as the objective function, the combination of the prediction and the control horizon of the LTV-MPC algorithm under different working conditions in terms of vehicle velocities and road adhesion coefficients are optimized offline with PSO. To ensure vehicle stability while tracking reference paths, a slip angle constraint is introduced to prevent tire forces from saturation. The lower layer is a RBFNN-PID steering angle tracking controller that generates electric current control signals for the steering motor. The nonlinear characteristics of the steering system are identified on-line with the RBFNN, and the PID controller’s control parameters are adjusted adaptively. The proposed hierarchical controller is validated on the CarSim-Matlab/Simulink simulation platform under double lane-changing conditions with various velocities and road adhesion coefficients. The effectiveness of the proposed strategy is verified through extensive simulation tests.

## 2. Overview

Figure 1 shows the diagram of the lateral control system utilizing the multi-layer controller this study proposes. The reference path is presented as given information.

The upper-layer LTV-MPC controller, whose parameters are optimized offline with the PSO algorithm, generates the target steering angle of the front wheel. The lower-layer RBFNN-PID controller outputs the electric current required by the steer-by-wire (SBW) system for rapid tracking of the target steering angle.

## 3. Vehicle Dynamics Model

This study adopts the “bicycle model” which has been widely used to develop the prediction model of MPC [30], as shown in Figure 2. The definitions of the parameters are listed in Table 1.

With Newton’s Second Law applied, a vehicle dynamics model can be built as given in Equation (1):(1)$$\left\{ \begin{array}{l} m{\dot{v}}_{y}=-mv_{x}w+2(F_{c}{}_{f}\cos(\delta_{f})+F_{l}{}_{f}\sin(\delta_{f}))+2F_{c}{}_{r}\\ \dot{\varphi }=w\\ I_{z}\dot{w}=2a(F_{c}{}_{f}\cos(\delta_{f})+F_{l}{}_{f}\sin(\delta_{f}))-2bF_{c}{}_{r}\\ \dot{Y}=v_{x}\sin(\varphi )+v_{y}\cos(\varphi ) \end{array}\right.$$
where *v_y_* and *v_x_* represent the vehicle longitudinal speed, and lateral speed in the body-fixed coordinate system respectively. *Y* is the vehicle lateral position in the Cartesian coordinate system.

To compute the longitudinal forces and lateral forces of the front and rear wheels with tire dynamics coupling considered, a semi-empirical nonlinear model, the Pacejka tire model, is adopted. As given in Equation (2), the tire longitudinal and lateral forces are described as nonlinear functions of their respective parameters: the slip angle α and the longitudinal slip ratio κ with the effect of vertical load *F_z_*, and the road friction coefficient *μ*. Figure 3 shows an example of the tire lateral forces versus longitudinal slip and slip angle, for the fixed values of *μ*.
(2)$$\left\{ \begin{array}{l} F_{l}{}_{f}=f_{lf}(\kappa_{f},\alpha_{f},\mu ,F_{zf})\\ F_{c}{}_{f}=f_{cf}(\kappa_{f},\alpha_{f},\mu ,F_{zf})\\ F_{l}{}_{r}=f_{lr}(\kappa_{r},\alpha_{r},\mu ,F_{zr})\\ F_{c}{}_{r}=f_{cr}(\kappa_{r},\alpha_{r},\mu ,F_{zr}) \end{array}\right.$$

We assume that the vehicle is equipped with an antilock brake system, and the tire longitudinal slip ratio *κ_f_* and *κ_r_* is therefore provided. As suggested in [22], if the slip angle is relatively small, it can be estimated with the vehicle longitudinal speed *v_y_*, the lateral speed *v_x_*, the Yaw rate *w*, and the front wheel steering angle *δ_f_*, as given in Equations (3) and (4):(3)$$\alpha_{f}=\frac{v_{y}+aw}{v_{x}}-\delta_{f}$$
(4)$$\alpha_{r}=\frac{v_{y}-bw}{v_{x}}$$
and the vertical load of the front and the rear wheels are defined as follows:(5)$$F_{zf}=\frac{bmg}{2(a+b)},\;F_{zr}=\frac{amg}{2(a+b)}$$

The nonlinear vehicle dynamics model described in Equations (1)–(5) can be rewritten in a compact form as defined below:(6)$$\dot{\xi }(t)=f_{\mu (t),\kappa (t)}(\xi (t),\delta_{f}(t))$$
where *ξ* is the state vector and *ξ* = [*v_y_,w*,*φ*,*Y*]*^T^*.

As we focus on the lateral control of the autonomous vehicle in this study, the longitudinal speed *v_x_* is set to be constant. For parameters in *ξ*, the yaw rate *w*, the heading angle *φ*, and the vehicle lateral position in Cartesian coordinate system *Y* can be measured with the yaw rate sensor and GPS/inertial measurement unit (IMU). Measuring the lateral velocity *v_y_* is more challenging both economically and technically. Instead, previous studies have proposed to estimate the vehicle sideslip angle, based on which *v_y_* can be calculated with *v_x_* being a known constant, utilizing vehicle dynamics models and observation techniques based on easily measurable parameters. Measurements from the differential GPS receiver and the IMU are fused with a Kalman filter to calculate the position, the orientation, and the longitudinal and lateral velocities of the vehicle [24,26]. For more adaptive and accurate estimation of the vehicle sideslip angle, measurable parameters such as the wheel angular velocity, yaw rate, and wheel angle are used [31,32,33,34]. Therefore, we draw the conclusion that all the parameters of vehicle states in *ξ* can be either directly measured or estimated from measurable parameters.

## 4. The PSO-LTV-MPC Controller

Both the prediction horizon and control horizon are optimized with PSO with regard to different speeds and road adhesion coefficients. In addition, a slip angle constraint, estimated based on the vehicle states and control variables, is imposed to avoid saturation of the lateral forces of tires and to ensure that the tires provide additional lateral forces when needed to resist the interference of external lateral forces.

### 4.1. Design of the LTV-MPC Controller

The objective of designing the LTV-MPC controller is to eliminate the deviations between the predicted vehicle outputs and their references with the optimal steering angles. We use the vehicle dynamics model described in Section 3 as the prediction model, to predict the vehicle outputs. A series of optimal steering angles are then calculated by solving the multi-objective multi-constraint QP problem to eliminate the deviations.

To obtain a finite-dimensional control problem, we discretize the nonlinear vehicle dynamics model in Equation (6) with a fixed sampling time *T_s_*:(7)$$\chi (d+1)=f(\chi (d),\Delta \delta_{f}(d))$$
(8)$$\chi (d)={\left[\begin{array}{cc} \xi (d) & \delta_{f}(d-1) \end{array}\right]}^{T}$$
where the control increment Δ*δ_f_* is chosen as the input, and Δ*δ_f_* (d) = *δ_f_* (d) − *δ_f_* (d−1).

The heading angle *φ* and the lateral position *Y* are chosen as the outputs *λ*. The new discrete LTV model is obtained by linearizing Equation (7) around an operating point with the method of state trajectory [24]:(9)$$\chi (d+1)=M_{d}\chi (d)+N_{d}\Delta \delta_{f}(d)$$
(10)$$\lambda (d)=H_{c}\cdot \chi (d)$$
where
(11)$$M_{d}=\left[\begin{array}{cc} A_{k} & B_{k}\\ 0_{1\times 4} & I \end{array}\right],\;N_{d}=\left[\begin{array}{c} B_{k}\\ I \end{array}\right],\;H_{c}=\left[\begin{array}{ccccc} 0 & 0 & 1 & 0 & 0\\ 0 & 0 & 0 & 1 & 0 \end{array}\right]$$

We define the current time as *t* and the measurable current vehicle states as *χ*(*t*). According to the control law of MPC, the control inputs remain unchanged between prediction horizon *C* and control horizon *P*. At each time step, the optimization problem of LTV MPC can be formed as follows:(12)$$\begin{array}{l} \underset{\Delta \delta_{f}(t)}{\min}{\sum_{i=1}^{P}\|Q\left[\lambda_{t}(t+i)-\lambda_{t,ref}(t+i)\right]\|}^{2}+{\sum_{i=0}^{C-1}\|R\left[\Delta \delta_{f,t}(t+i)\right]\|}^{2}\\ subj.\;to\;\mathrm{Eq}.(12-13) \end{array}$$
(13)$$\begin{array}{l} \chi_{t}(d+1)=M_{d}\chi_{t}(d)+N_{d}\Delta \delta_{f,t}(d)\\ \lambda_{t}(d)=H_{c}\cdot \chi_{t}(d) \end{array}$$
(14)$$\Delta \delta_{f,t}(d)=\delta_{f,t}(d)-\delta_{f,t}(d-1)$$
(15)$$\begin{array}{l} \delta_{f,\min}\leq \delta_{f,t}(d)\leq \delta_{f,\max}\\ \Delta \delta_{f,\min}\leq \Delta \delta_{f,t}(d)\leq \Delta \delta_{f,\max} \end{array}$$
where *λ_ref_* = [*φ_ref_, Y_ref_*]*^T^* represents the reference signal. *Q*, *R* represents the weight matrix of controlled outputs and inputs respectively. The prediction model in Equations (13) and (14) is used to predict the vehicle’s behavior. The inequation in Equation (15) are the hard constraints of the control input and the control increment.

The above optimization problem is transformed into the following QP form:(16)$$\begin{array}{l} \underset{\Delta U(d)}{\min}X^{T}\tilde{H}X+GX^{T}\\ subj.\;to\left[\begin{array}{c} I\\ -I\\ R\\ -R \end{array}\right]X\leq \left[\begin{array}{c} \Delta \delta_{f,\max}\times ones(C,1)\\ -(\Delta \delta_{f,\min}\times ones(C,1))\\ \left(\delta_{f,\max}-\delta_{f}(d-1)\right)\times ones(C,1)\\ \left(-\delta_{f,\min}+\delta_{f}(d-1)\right)\times ones(C,1) \end{array}\right] \end{array}$$
where
(17)$$\begin{array}{cc} \begin{array}{c} X=\Delta U(d)\\ \tilde{H}=2\Xi_{d}{}^{T}Q\Xi_{d}+R\\ E_{d}=\Gamma_{d}\chi (d)-Y_{ref}\\ G=2E_{d}{}^{T}Q\Xi_{d} \end{array} & Y_{ref}=\left[\begin{array}{c} \lambda_{ref}(d+1)\\ \lambda_{ref}(d+2)\\ \vdots \\ \lambda_{ref}(d+P) \end{array}\right]\Delta U(d)=\left[\begin{array}{c} \Delta \delta_{f}(d)\\ \Delta \delta_{f}(d+1)\\ \vdots \\ \Delta \delta_{f}(d+C) \end{array}\right]R={\left[\begin{array}{cccc} 1 & 0 & \cdots & 0\\ 1 & 1 & \ddots & \vdots \\ \vdots & \vdots & \ddots & 0\\ 1 & 1 & \cdots & 1 \end{array}\right]}_{C\times C}\\ \Gamma_{d}=\left[\begin{array}{c} H_{c}M_{d}\\ H_{c}M_{d}M_{d+1}\\ \vdots \\ H_{c}\prod_{i=0}^{P-1}M_{d+i} \end{array}\right] & \Xi_{d}=\left[\begin{array}{cccc} H_{c}N_{d} & 0 & \cdots & 0\\ H_{c}M_{d}N_{d} & H_{c}N_{d} & \cdots & 0\\ \vdots & \vdots & \ddots & \vdots \\ H_{c}\prod_{i=1}^{P-1}M_{d+i}N_{d} & H_{c}\prod_{i=2}^{P-1}M_{d+i}N_{d+1} & \cdots & H_{c}\prod_{i=C}^{P-1}M_{d+i}N_{d+C-1} \end{array}\right] \end{array}$$

By solving the QP problem presented in Equation (16) at time *t*, we get the control increment sequence Δ*U_t_*^*^ = [Δ*δ_f_*_,*t*_^*^,…,Δ*δ_f_*_,*t*+*C*_^*^ ]*^T^*. And the optimal control law is obtained:(18)$$\delta_{f}(t)=\Delta \delta_{f,t}{}^{*}+\delta_{f}(t-1)$$
where Δ*δ_f_*_,*t*_^*^ is the first element of Δ*U_t_*^*^.

It should be noted that no constraint on the slip angle is so far imposed when solving the aforementioned objective function, thus the control sequence obtained is inapplicable under limited working conditions.

A discrete LTV prediction model of the tire slip angle is obtained by discretizing and linearizing the Equations (3) and (4):(19)$$\alpha (d)=E_{\alpha }\chi (d)+Z_{\alpha }\Delta \delta_{f}(d)$$
where
(20)$$\alpha ={\left[\alpha_{f},\alpha_{r}\right]}^{T},\;Z_{\alpha }={\left[-1,0\right]}^{T},E_{\alpha }=\left[\begin{array}{ccccc} \frac{1}{v_{x}} & \frac{a}{v_{x}} & 0 & 0 & -1\\ \frac{1}{v_{x}} & \frac{-b}{v_{x}} & 0 & 0 & 0 \end{array}\right]$$

Based on the measurable current vehicle states *χ*(*t*), the predictive outputs of tire slip angle over the prediction horizon are obtained:(21)$$\alpha_{t}(d+1)=E_{\alpha }\chi_{t}(d+1)+Z_{\alpha }\Delta \delta_{f,t}(d+1)\;$$

We restraint the slip angle in the region as defined below
(22)$$\alpha_{\min}\leq \alpha_{t}(d+1)\leq \alpha_{\max}$$

Then, the constraint of the slip angle is established as follows
(23)$$\left[\begin{array}{c} \Omega_{\alpha }\\ -\Omega_{\alpha } \end{array}\right]X\leq \left[\begin{array}{c} \alpha_{\max}\times ones(C,1)-S_{\alpha }\chi (d)\\ S_{\alpha }\chi (d)-\alpha_{\min}\times ones(C,1) \end{array}\right]$$
where
(24)$$S_{\alpha }=\left[\begin{array}{c} E_{\alpha }M_{d}\\ E_{\alpha }M_{d}M_{d+1}\\ \vdots \\ E_{\alpha }\prod_{i=0}^{P-1}M_{d+i} \end{array}\right]\Omega_{\alpha }=\left[\begin{array}{cccc} E_{\alpha }N_{d}+Z_{\alpha } & 0 & \cdots & 0\\ E_{\alpha }M_{d}N_{d}+Z_{\alpha } & E_{\alpha }{\tilde{B}}_{k}+Z_{\alpha } & \cdots & 0\\ \vdots & \vdots & \ddots & \vdots \\ E_{\alpha }\prod_{i=1}^{P-1}M_{d+i}N_{d}+Z_{\alpha }{}_{\alpha } & E_{\alpha }\prod_{i=2}^{P-1}M_{d+i}N_{d+1}+Z_{\alpha } & \cdots & E_{\alpha }\prod_{i=C}^{P-1}M_{d+i}N_{d+C-1}+Z_{\alpha } \end{array}\right]$$

Lastly, by solving the QP problem in Equation (16) with the slip angle constraint in Equation (23), the control input increment is produced with the stability of the vehicle considered.

### 4.2. Optimizing Controller Parameters Using PSO Algorithm

To further improve the LTV-MPC controller’s performance, the PSO algorithm is adopted to find the optimal pairs of prediction horizon and control horizon with varying vehicle speed *V* and road friction coefficient *μ*.

As the average lateral error is chosen as the evaluation index of the path tracking performance, the fitness function of PSO is defined to be the RMS of the error in lateral positions, as expressed in Equation (25).
(25)$$f=\sqrt{\frac{1}{N}\sum_{i=1}^{N}{\left(Y(i)-Y_{ref}(i)\right)}^{2}}$$

To achieve a tradeoff between the local and global searching ability of PSO, the inertia weight *W* of updating particle velocity is linearly decreased.
(26)$$W(f)=W_{\max}-(W_{\max}-W_{\min})\frac{f}{T_{\max}}$$
where *T*_max_ represents the maximum iteration number. The maximum and minimum inertia weight is set empirically to 0.9, 0.4 respectively.

As shown in Table 2, both prediction horizon and control horizon are optimized with PSO when the vehicle speed *V* = [10,15,20,25] m/s, and the friction coefficient of road *μ* = [0.3, 0.8]. The optimization process involves 4 steps:Step 1:Initialize particle positions and velocities. Set the population number *N_f_* = 30, the maximum iteration number *k* = 30, learning factor *C_1_* = *C_2_* = 2, particle position information *P_p_* = [*P*, *C*]. The upper boundary of the particle’s position is set to [30,30] while the lower boundary is set to [1,1]. The particle velocity *V_p_* ϵ [–1,1]. The current iteration number *T* = 1;Step 2:Compute the particle’s fitness *f*, and choose the individual extremum *P_I_*_, *b*_ and the group extremum *P_G_*_, *b*_;Step 3:Update *P_p_*, *V_p,_* and *f* based on *P_I_*_,*b*_ and *P_G_*_, *b*_ obtained in Step 2, and then update the *P_I_*_,*b*_ and *P_G_*_, *b*_;Step 4:Check to see if the termination condition is satisfied. If not, back to step 2; else the procedure terminates.

## 5. The RBFNN-PID Front Wheel Angle Tracking Controller

With the aim to design a control strategy that is capable of handling the nonlinear time-varying (NTV) characteristics of the SBW system and outputs control signals that are executable by steering actuators, a RBFNN-PID controller is built as the lower-layer controller. First, a steering system model that depicts the electromechanical connection between the actual steering angle and its target is established. Then, the RBF neural network, which has fast learning convergence and strong nonlinear fitting ability, is implemented to identify the characteristics of the SBW system and tune the PID controller’s control parameters on-line.

### 5.1. Modeling the SBW System

The structural of the SBW model is shown in Figure 4, which is composed of an electric motor, a rack, a steering trapezium, and directive wheels. An SBW model that models the dynamics of the electric motor and steering actuators is built. We assume that the left steering angles of front wheels and the right one is identical. We derive the following equations:

The dynamics model of the electric motor is expressed as
(27)$$\left\{ \begin{array}{l} J_{s}{\ddot{\theta }}_{s}+B_{s}{\dot{\theta }}_{s}=T_{s}-T_{\delta }\\ T_{\delta }=K_{s}(\theta_{s}-\frac{x_{r}g_{s}}{r_{p}})\\ T_{s}=K_{i}i \end{array}\right.$$

The rack dynamics model is expressed as
(28)$$m_{r}{\ddot{x}}_{r}+B_{r}{\dot{x}}_{r}+K_{r}x_{r}=\frac{K_{s}(\theta_{s}-\frac{x_{r}g_{s}}{r_{p}})}{r_{p}}\eta_{p}-\frac{2\left(\frac{K_{kp}x_{r}g_{s}}{N_{l}}-\delta \right)}{N_{l}}$$

The dynamics model of the directive wheel is expressed as
(29)$$J_{fw}\ddot{\delta }+B_{kp}\dot{\delta }=K_{kp}(\frac{x_{r}}{N_{l}}-\delta )-T_{f}$$

Definitions and values of variables presented in Equations (27)–(29) are shown in Table 3.

The steering resistant torque of the front wheel *T_f_* is composed of the front wheel rolling resistant torque *M_1_* and self-aligning torque *M_2_*, as expressed in Equation (30):(30)$$T_{f}=M_{1}+M_{2}=\frac{fcbmg}{a+b}+\frac{cbmg\cos\delta }{180(a+b)}\sin2\beta +\frac{bmv_{x}{}^{2}r}{(a+b)R}\sin\gamma \cos\delta$$
where *c*, *β,* and *γ* represent the offset distance of the front pin, the kingpin inclination, and the caster angle, respectively. *R* denotes the turning radius of the vehicle, and *f* denotes the rolling resistance coefficient.

### 5.2. The Steering Angle Tracking Controller Based on RBFNN-PID

Figure 5 illustrates the architecture of the proposed RBFNN-PID controller. The PID controller’s input is the deviation of the actual steering angle from its target, while the output is the current to be executed by the electric motor. Three parameters *k_P_*, *k_I_*, *k_D_* of the PID controller are tuned adaptively with the RBFNN [35,36].

The incremental PID controller is adopted and the control error is defined below:(31)$$d_{k}=\delta_{f,k}-\delta_{k}$$

The inputs of the controller are:
(32)$$\{\begin{array}{l} I_{1}=d_{k}-d_{k-1}\\ I_{2}=d_{k}\\ I_{3}=d_{k}-2d_{k-1}+d_{k-2} \end{array}$$

The incremental PID control low is expressed as follows:(33)$$\{\begin{array}{l} i_{k}=i_{k-1}+\Delta i_{k}\\ \Delta i_{k}=(K_{P0}+\Delta K_{P})I_{1}+(K_{I0}+\Delta K_{I})I_{2}+(K_{D0}+\Delta K_{D})I_{3} \end{array}$$
where *i_k_* represents the PID controller’s output at time *k*, and *K_P_*_0_, *K_I_*_0_, *K_D_*_0_ denote the corresponding initial values of *K_P_*, *K_I_*, *K_D_* respectively.

A RBFNN is implemented to identify the characteristics of the SBW system through analyzing the its sensitivity between the input signal and the output variable. Then, the control parameters of PID is tuned adaptively with gradient descent.

The index of tuning of the RBFNN is defined as follows:(34)$$E_{k}=\frac{1}{2}d_{k}{}^{2}$$

To minimize *E_k_*, the incremental parameters of PID is tuned with gradient descent:(35)$$\left\{ \begin{array}{c} \Delta K_{P}=-\tau \frac{\partial E}{\partial K_{P}}=-\tau \frac{\partial E}{\partial \delta }\frac{\partial \delta }{\partial \Delta i}\frac{\partial \Delta i}{\partial K_{P}}=\tau d(t)\frac{\partial \delta }{\partial \Delta i}I_{1}\\ \Delta K_{I}=-\tau \frac{\partial E}{\partial K_{I}}=-\tau \frac{\partial E}{\partial \delta }\frac{\partial \delta }{\partial \Delta i}\frac{\partial \Delta i}{\partial K_{I}}=\tau d(t)\frac{\partial \delta }{\partial \Delta i}I_{2}\\ \Delta K_{D}=-\tau \frac{\partial E}{\partial K_{D}}=-\tau \frac{\partial E}{\partial \delta }\frac{\partial \delta }{\partial \Delta i}\frac{\partial \Delta i}{\partial K_{D}}=\tau d(t)\frac{\partial \delta }{\partial \Delta i}I_{3} \end{array}\right.$$
where *τ* represents the learning efficiency which falls in the interval [0,1], and *∂δ*/*∂*Δ*i* is the sensitivity of the controlled object, which is identified with the RBFNN.

The adopted RBFNN is a three-layer feedforward network, and its structure is shown in Figure 6. *X* denotes the input vector of the network which consists of the PID controller’s output incremental current Δ*i_k_*, the actual steering angle at the present time *δ_k_*, and at the previous time *δ_k_*_−1_.

The output of the RBFNN *δ_m_*_,*k*_, is computed as follows:(36)$$\delta_{m,k}=\sum_{j=1}^{m}w_{j}h_{j}$$
where *w_j_* represents the connection weight between *h_j_* and *δ_m_*_,*k*_, and the transfer function *h_j_* is calculated as in Equation (36).
(37)$$h_{j}=\exp\left[-\frac{{\left\|X-c_{j}\right\|}^{2}}{2b_{j}^{2}}\right]$$
where *c* = [*c_j_*_1_, *c_j_*_2_, …, *c_jn_*]*^T^*, *b* = [*b*_1_, *b*_2_, …, *b_m_*]*^T^* represent the center vector and the base width vector of the *j^th^* neuron, respectively.

Finally, the aforementioned sensitivity of the controlled object is identified as
(38)$$\frac{\partial \delta_{k}}{\partial \Delta i_{k}}\approx \frac{\partial \delta_{m}{}_{,k}}{\partial \Delta i_{k}}=\sum_{i=1}^{m}w_{i}h_{i}\frac{c_{ij}-x_{i}}{b_{i}{}^{2}}$$

## 6. Simulation and Results

### 6.1. Simulation Design

The double lane change (DLC) trajectory proposed in [24] is used as the reference trajectory to and a series of simulation studies is conducted jointly with CarSim-Matlab/Simulink to verify the proposal of this study, A B-Class Hatchback model in CarSim is used and its parameters are shown in Table 4. The RMS value of the error in lateral positions is adopted to evaluate the 4 variants of controllers’ performance. Controller A is the pure-pursuit controller embedded in the CarSim driver model, controller B is the upper-level LTV-MPC controller presented in Section 4.1, controller C is the proposed two-layer tracking controller without the slip angle constraint, and controller D is the proposed two-layer tracking controller with the slip angle constraint.

Three test scenarios are implemented. Scenario 1 presents an emergency obstacle avoidance scene on a high adhesion road, in which the performance is compared among the controller A, B, and C. Scenario 2 presents an emergency obstacle avoidance scene on a low adhesion road, in which the performance is compared among the controller A, B, and C. Scenario 3 presents an emergency obstacle avoidance scene on a low adhesion road, in which the performance is compared between the controller C and D.

### 6.2. Analysis of Simulation Results

#### 6.2.1. Scenario 1

On the road surface which friction coefficient is 0.8, three controllers are tested at different vehicle speeds ranging from 10 m/s to 25 m/s. Figure 6 show the tracking performance of 3 controllers. The lateral tracking deviations of 3 controllers are given in Table 5.

As shown in Figure 7, controller B and controller C both perform well at all speeds, while controller A can only achieve good tracking performance at relatively low speed (10 m/s). As the velocity increases, the lateral tracking deviations of controller B and controller C increase slowly, while the lateral tracking deviation of controller A increases significantly and gets much higher than another two MPC-based controllers. Besides, at the speed of 25 m/s, both controller B and controller C can still help to guarantee the vehicle’s lateral stability while tracking the reference path smoothly and accurately.

Moreover, the performances of controller B and controller C are quite close as shown in Table 5, which indicates that the lower-level RBFNN-PID controller proposed in Section 5 can effectively deal with the NTV characteristics of the SBW system and hence achieve no worse performance in path tracking as compared to the case of controller B where the steering system is completely neglected.

#### 6.2.2. Scenario 2

To test the tracking performance of each controller on slippery roads, we set the road friction coefficient to 0.3 in this scenario. The lateral tracking deviations of 3 controllers are given in Table 6.

Due to the limited amount of frictional force available, the tracking error of all controllers in this scenario increases as the vehicle speed increases, as shown in Figure 8. However, it’s observed from simulation results that controller B and controller C still manage to track the path at high accuracy while maintaining the stability of the vehicle at medium and low speeds, and also achieve acceptable path tracking performance at high speeds.

#### 6.2.3. Scenario 3

In this scenario, the tracking performance of controller C and controller D on a low friction coefficient road (*μ* = 0.3) is examined. The difference between the two controllers is that controller D is imposed with the slip angle constraint. Figure 9, Figure 10 and Figure 11 compare these two controllers in tracking performance. The lateral tracking deviations of both controllers are shown in Table 7.

As shown in Figure 10, when the vehicle speed is at or above 10 m/s, the front wheel steering angle of the vehicle with controller D is within a considerably smaller range as compared to the vehicle with controller C, indicating that the vehicle is more stable with controller D at high speeds. This is because the amount of frictional force available is greatly reduced and can be easily saturated with low friction coefficient conditions. The nonlinear tire model without the slip angle constraint lose the ability to predict the lateral tire force when the frictional tire forces saturates. With external lateral force applied at this time, the vehicle loses stability. On the contrary, the controller constrained by the slip angle ensures that the tires provide additional lateral forces when needed to resist the interference of external lateral forces such as lateral wind, thereby improving vehicle stability on low friction coefficient roads. Moreover, the results given in Table 7 and shown in Figure 9 suggest that the tracking performance of controller D matches with that of controller C when the vehicle speed is relatively low and outperforms when the vehicle speed is above 20 m/s. It confirms that imposing the slip angle constraint does not affect the tracking performance while guaranteeing the vehicle stability.

In conclusion, simulation results show that the proposed two-layer path tracking controller can track the reference paths accurately and smoothly while ensuring the vehicle’s stability, and is robust to dynamic changes in road surface conditions and vehicle speeds.

## 7. Conclusions

A hierarchical controller for lateral path tracking control of autonomous vehicles is proposed. The upper layer is a PSO-LTV-MPC controller considering the slip constraint which is estimated from measurable vehicle states, to avoid saturation of the lateral forces of tires. The prediction and the control horizon are both optimized offline with PSO, thereby improving the performance of the controller under dynamic conditions. The lower layer is a RBFNN-PID controller, which identifies the nonlinear characteristics of the steering system online and outputs an executable control signal for actuators, that is, the current of the steering executive motor, to achieve rapid tracking of the target steering angle produced by the upper controller. We verify the effectiveness of the proposed approaches with CarSim-Matlab/Simulink joint simulations under double lane changing conditions with the vehicle velocity and the road friction coefficient varies. The results show that as compared to the competitors, the proposed controller tracks the target path accurately while maintaining a good level of vehicle stability, and is robust to changes in road attachment conditions and vehicle speeds. Future research will attempt to further improve the tracking performance with coordinated lateral and longitudinal control.

## Figures and Tables

**Figure 1 sensors-20-03689-f001:**
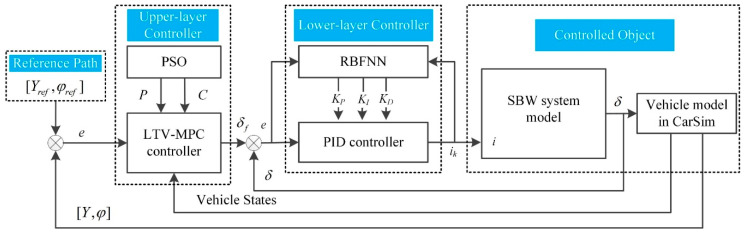
The diagram of the path tracking control system.

**Figure 2 sensors-20-03689-f002:**
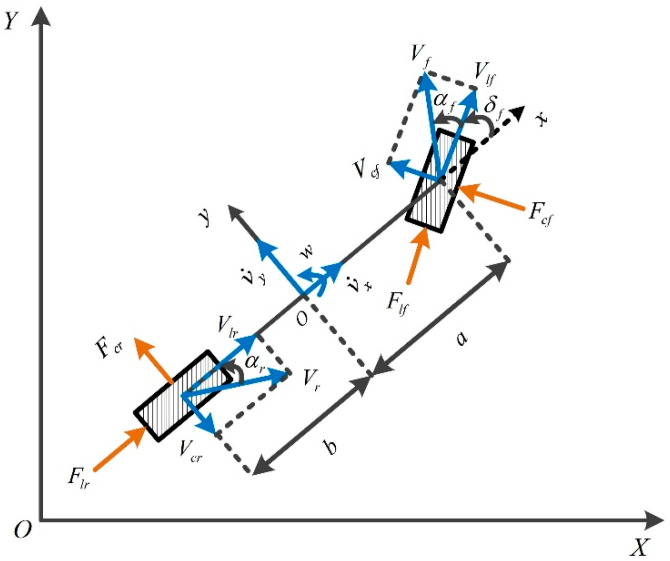
Vehicle dynamics model.

**Figure 3 sensors-20-03689-f003:**
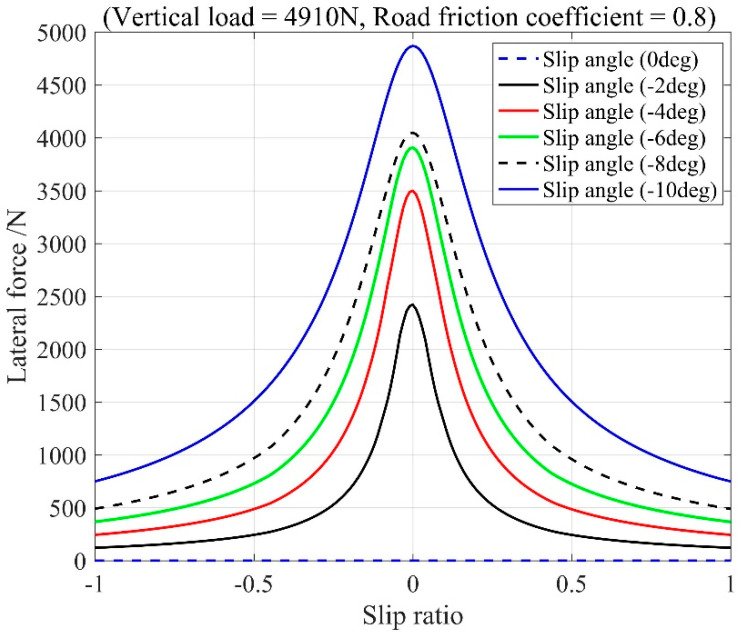
The tire lateral forces versus longitudinal slip and slip angle.

**Figure 4 sensors-20-03689-f004:**
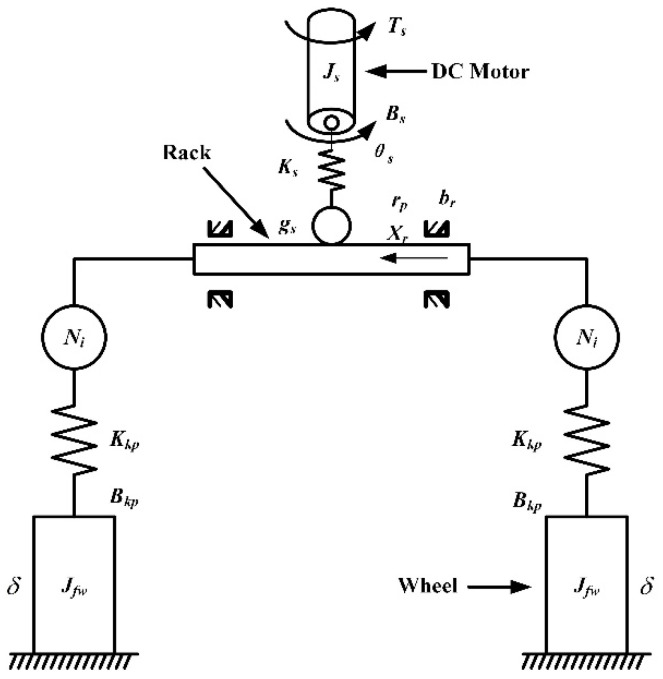
SBW system model.

**Figure 5 sensors-20-03689-f005:**
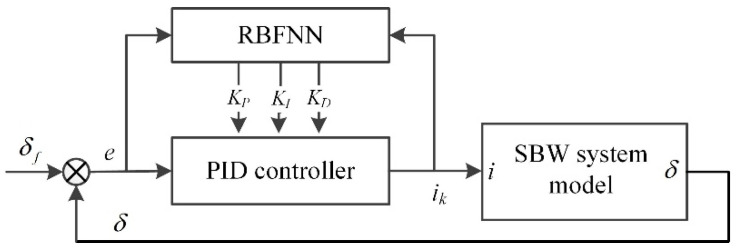
The architecture of the RBFNN-PID controller.

**Figure 6 sensors-20-03689-f006:**
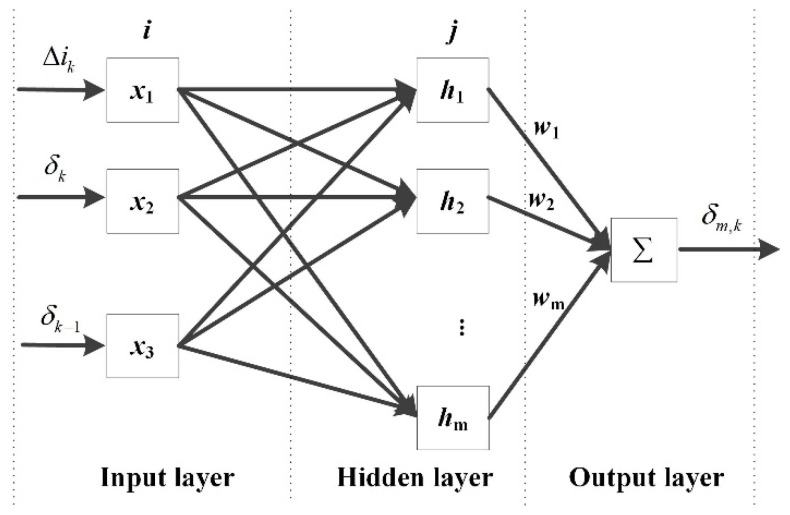
The structure of RBFNN.

**Figure 7 sensors-20-03689-f007:**
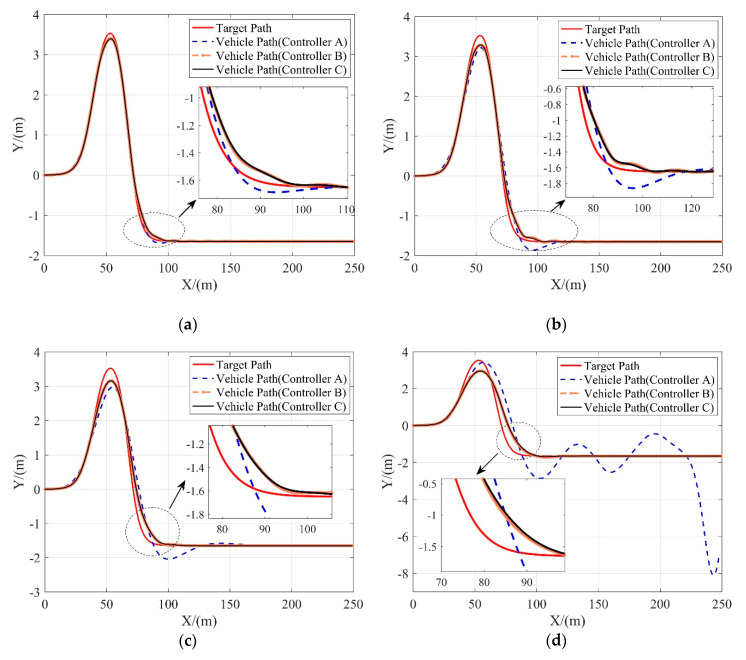
Tracking performance of 3 controllers in terms of the lateral position at different vehicle velocities. (**a**) 10 m/s; (**b**) 15 m/s; (**c**) 20 m/s; (**d**) 25 m/s.

**Figure 8 sensors-20-03689-f008:**
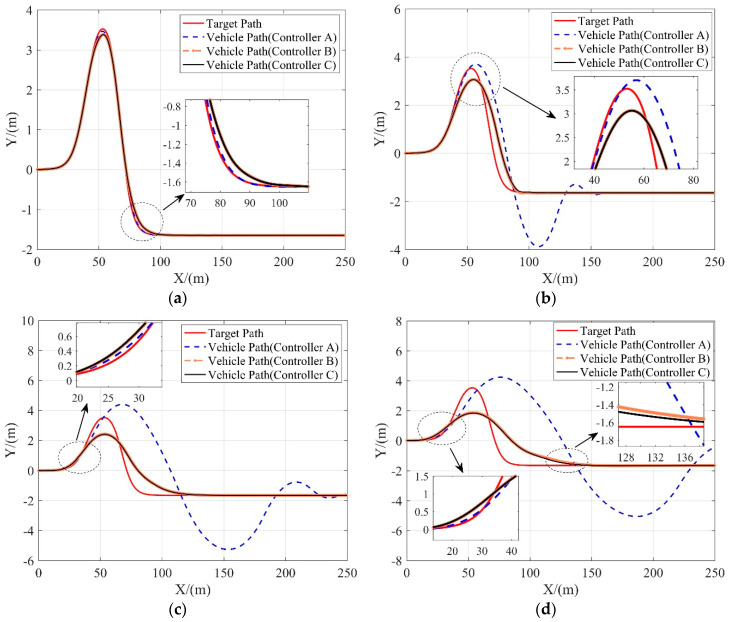
Tracking performance of 3 controllers in terms of the lateral position at different vehicle velocities. (**a**) 10 m/s; (**b**) 15 m/s; (**c**) 20 m/s; (**d**) 25 m/s.

**Figure 9 sensors-20-03689-f009:**
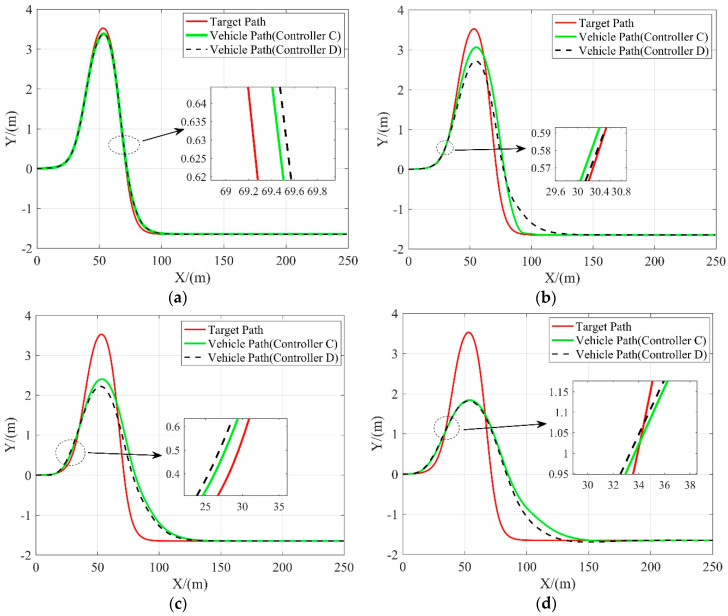
Tracking performance of 3 controllers in terms of the lateral position at different vehicle velocities. (**a**) 10 m/s; (**b**) 15 m/s; (**c)** 20 m/s; (**d**) 25 m/s.

**Figure 10 sensors-20-03689-f010:**
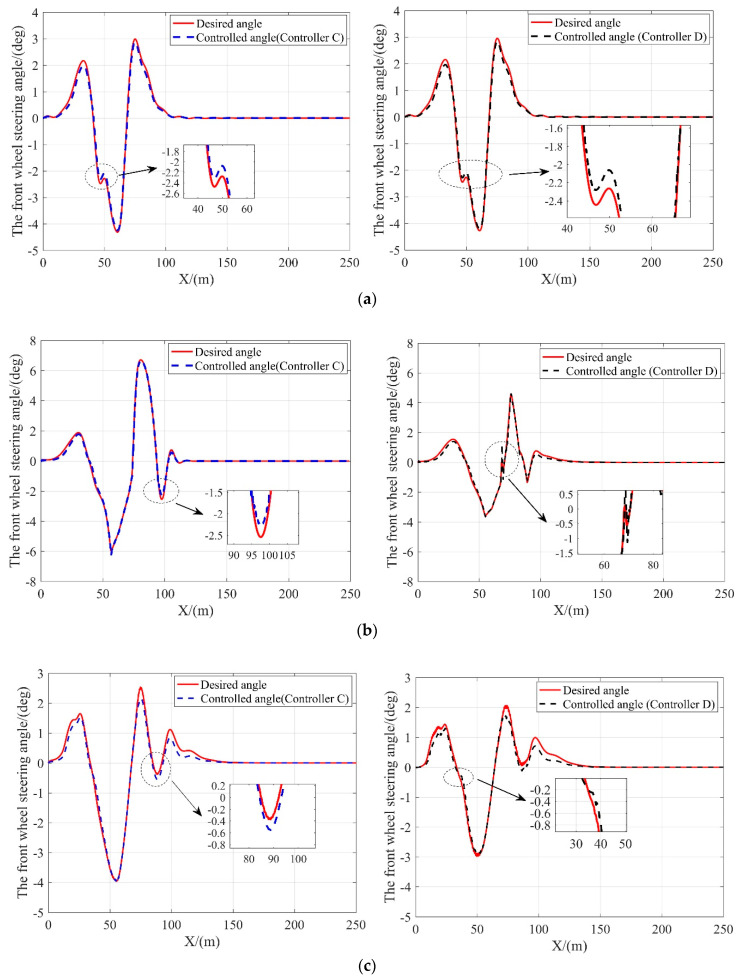
The desired and controlled steering angle of the front wheel with the controller C and D at different vehicle velocities. (**a**) 10 m/s; (**b**) 15 m/s; (**c**) 20 m/s; (**d**) 25 m/s.

**Figure 11 sensors-20-03689-f011:**
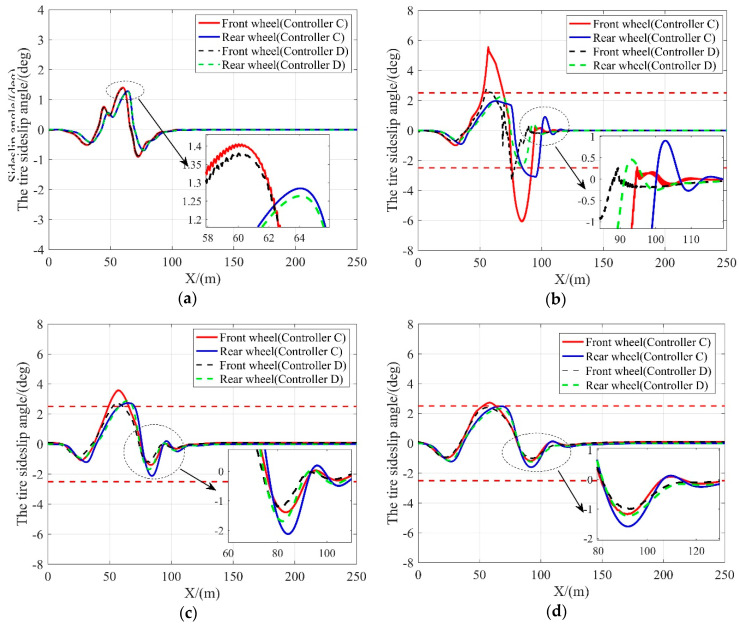
Slip angle at different vehicle velocities. (**a**) 10 m/s; (**b**) 15 m/s; (**c**) 20 m/s; (**d**) 25 m/s.

**Table 1 sensors-20-03689-t001:** Bicycle model parameters.

Symbol	Description
*F_lf_*/*F_cf_*	Longitudinal force/lateral force of the front wheel
*F_lr_*/*F_cr_*	Longitudinal force/lateral force of the rear wheel
*a*/*b*	Distance of front/rear axle from the center of gravity
*m*	Vehicle Mass
*r*	Wheel radius
*κ_f_*/*κ_r_*	The tire longitudinal slip ratio of front/rear tires
*C_cr_/C_lr_*	Lateral stiffness/longitudinal stiffness of rear tire
α*_f_*/α*_r_*	Slip angle of front/rear tire
*C_cf_*/*C_lf_*	Lateral stiffness/longitudinal stiffness of front tire
*I_z_*	Vehicle inertia
*δ_f_*	Front wheel steering angle
*φ*	Heading angle
*w*	Yaw rate

**Table 2 sensors-20-03689-t002:** Optimized pairs of prediction horizon and control horizon.

Vehicle Speed(m/s)	The Friction Coefficient of Road	Prediction Horizon *N_P_*	Control Horizon *N_C_*
10	0.3	8	7
	0.8	8	8
15	0.3	11	2
	0.8	8	7
20	0.3	23	6
	0.8	9	9
25	0.3	25	2
	0.8	10	10

**Table 3 sensors-20-03689-t003:** Parameters of the SBW model.

Symbol	Definition	Value
*J_s_*	Motor shaft inertia	0.00019 kg m^2^
*B_s_*	Motor shaft damping coefficient	0.0034 N m/(rad/s)
*K_s_*	Motor shaft torsional stiffness	115 N m/rad
*g_s_*	Motor speed reduction ratio	10
*m_r_*	Rack mass	2.57 kg
*B_r_*	Rack damping coefficient	314 N/(m/s)
*K_r_*	Rack spring constant	91085 N/m
*r_p_*	Pinion radius	0.009 m
*ŋ* *_p_*	Steering gear transmission efficiency	1
*J_fw_*	Steering tire inertia	0.82 kg m^2^
*K_i_*	Torque coefficient of the motor	0.0718 N m/A
*N_l_*	Transmission ratio from rack to steering tire	0.1003
*B_kp_*	Steering tire damping coefficient	197 N m/(rad/s)
*K_kp_*	Steering tire torsion stiffness	39951.6 N m/rad

**Table 4 sensors-20-03689-t004:** Vehicle model parameters.

Symbol	Value
*m*	1843 kg
*I_z_*	4175 kg m^2^
*a*	1.232 m
*b*	1.468 m
*r*	0.308 m

**Table 5 sensors-20-03689-t005:** Lateral deviations of 3 controllers at different vehicle velocities *V*(m/s).

Controller	$\Delta Y_{RMS}\;(m)$			
	*V* = 10	*V* = 15	*V* = 20	*V* = 25
A	0.0672	0.1608	0.2835	1.6067
B	0.0546	0.0973	0.1643	0.2964
C	0.0584	0.0985	0.1679	0.3022

**Table 6 sensors-20-03689-t006:** Lateral deviations of 3 controllers at different vehicle velocities *V*(m/s).

Controller	$\Delta Y_{RMS}\;(m)$			
	*V* = 10	*V* = 15	*V* = 20	*V* = 25
A	0.0320	0.8794	2.2412	2.6585
B	0.0620	0.3348	0.4776	0.6731
C	0.0622	0.3365	0.4764	0.6657

**Table 7 sensors-20-03689-t007:** Lateral deviations of 2 controllers at different vehicle velocities *V*(m/s).

Controller	$\Delta Y_{RMS}\;(m)$			
	*V* = 10	*V* = 15	*V* = 20	*V* = 25
C	0.0622	0.3365	0.4764	0.6657
D	0.0663	0.3609	0.4616	0.6229

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
