# Peer review of "A Two-Layer Controller for Lateral Path Tracking Control of Autonomous Vehicles"

_sensors, 2020, doi:10.3390/s20133689_

Round 1

Reviewer 1 Report

The author suggest that a limitation of other contributions is the adoption of linear tyre models, which undoubtedly lose effectiveness for non-small tyre slip angles. However the use of the single-track (bicycle) model is also an approximation with important limitations.

Nonetheless, the authors mention that they use a Pacejka tyre model, but actually they are using a linear model, equation 2. So this would suffer from the same limitation discussed in the introduction.

Page1, line 39, introduction, please sort out missing content in line 39: "and ".

Typo in Table 1:"sidesilp". 

What is "Y" in equation 1?

No units are provided in Table 3.

It is unclear how the sideslip angle is estimated. This is a very challenging issue in the research community.

Table 4 suggests the same value for Ccf and Clf, hence the same behaviour for the tyre longitudinal and lateral interaction, which is far from reality, also considering that no combined interaction is considered.

Author Response

Dear Reviewer: Please see the attachement. Regards.

Reviewer 2 Report

This paper presents a two-layer controller for lateral path tracking control of highly automated vehicles with LTV-MPC and RBF-NN.

The proposed two-layer approach is interesting and the methods used in each layer are widely used schemes in automotive society. However, the author should improve some content.  

It is hard to see the difference between controller B and C. Please change the color or line style.

Please show both the desired steering wheel angle and controlled steering wheel angle. In Figure 9 (c) and (d), the front wheel angle shows oscillation at high speed. Oscillaton at the steering wheel angle can cause these unsuitable front tire slip angle.

Please specify the advantage of the proposed method. Many control algorithms are already developed and implemented in a real test vehicle.  (ex, Backstepping: Kim et al, "Nonlinear Backstepping Control Design for Coupled Nonlinear Systems under External Disturbances", Kang et al, “Observer-based backstepping control method using reduced lateral dynamics for autonomous lane-keeping system”, SMC, LQ, ...)

Please describe more details compared to not only MPC but also these control algorithms. The reason why the author uses MPC with NN should be improved.

Author Response

(The authors gave the same response as above.)

Round 2

Reviewer 1 Report

Thank you for providing the revised version of the paper.

All my concerns were addressed, except one:

- you kindly mention that equations 3 and 4 are used to estimate the "sideslip angle" but actually that would be the slip angles (or, in some books, called the "tyre sideslip angle"). My point was that estimating the vehicle sideslip angle (atan(vy/vx)) is difficult, which means that estimating vx and vy is difficult, and those parameters are used in equations 3 and 4. But how are vx and vy estimated? In a previous comment the authors suggest that "All 5 parameters can either be directly measured or estimated from measurable parameters", but again, a more detailed discussion would be beneficial. For example there are plenty of papers in the literature dealing with "vehicle sideslip angle estimation" which might be referred to.

Additional notes:

- there were inconsistencies between the lines indicated in the cover letter and those in the manuscript, for example in the cover letter the authors mention that "It’s worth noticing that a constraint on the sideslip angle is still required and the reason is presented in detail in Lines 84-89 in the revised manuscript" but, having checked, it seems to me that it should be lines 149-153

- note that the caption of Figure 3 erroneously includes part of the manuscript text

Author Response

Dear Reviewer:

Regards.

Reviewer 2 Report

The manuscript has been properly revised and supplemented.

Please add description the difference between the low-layer scheme in your paper and the paper "Electric power steering nonlinear problem based on proportional–integral–derivative parameter self-tuning of backpropagation neural network"

Author Response

Dear Reviewer,

Regards
